



# The general formulation for runoff components estimation and attribution at mean annual time scale

Yufen He[1], Changming Li[1*], Hanbo Yang[1*]

1 State Key Laboratory of Hydroscience and Engineering, Department of Hydraulic Engineering,

Tsinghua University, Beijing 100084, China

*Correspondence: Changming Li (licm_13@163.com), Hanbo Yang

(yanghanbo@tsinghua.edu.cn)

## Abstract

Estimating runoff components, including surface flow, baseflow and total runoff is essential for understanding precipitation partition and runoff generation and facilitating water resource management. However, a general framework to quantify and attribute runoff components is still lacking. Here, we propose a general formulation through observational data analysis and theoretical derivation based on the two-stage Ponce-Shetty model (named as the MPS model). The MPS model characterizes mean annual runoff components as a function of available water with one parameter. The model is applied over 662 catchments across China and the contiguous United States. Results demonstrate that the model well depicts the spatial variability of runoff components with $R^2$ exceeding 0.81, 0.44 and 0.80 for fitting surface flow, baseflow and total runoff, respectively. The model effectively simulates multi-year runoff components with $R^2$ exceeding 0.97, and the proportion of runoff components relative to precipitation with $R^2$ exceeding 0.94. By using this conceptual model, we elucidate the responses of surface flow and baseflow to available water and environmental factors for the first time. The surface flow is jointly controlled by precipitation and environmental factors, while baseflow is mainly influenced by environmental factors in most catchments. The universal and concise MPS model offers a new perspective on the long-term catchment water balance, facilitating broader application in large-sample investigations without complex parameterizations and providing an efficient tool to explore future runoff variations and responses under changing climate.



**Key Points**

(1) A general and concise formulation is proposed to quantify, and attribute mean annual
surface flow, baseflow and total runoff.
(2) The formulation characterizes runoff components as a function of available water without
additional and complicated parameter calculation.
(3) The formulation performs well in quantifying and attributing runoff components in 662
catchments.

**1. Introduction**

Runoff is the primary freshwater resource accessible for human life and plays an essential role
in the water cycle (He et al., 2022; Wang et al., 2024). Based on the propagation time and
hydraulic response of a catchment, total runoff ($Q$) can be divided into baseflow ($Q_b$) and surface
flow ($Q_s$) (Gnann et al., 2019; Singh et al., 2019). Baseflow, also referred to as slow flow, is
defined as the flow that originates from groundwater and other delayed sources (such as wetlands,
lakes, snow and ice), and generally sustains streamflow during dry periods (Gnann, 2021; Hall,
1968). Baseflow is the relatively stable component of runoff, playing a vital role in aquatic
ecosystems (de Graaf et al., 2019; Price et al., 2011), water quality (Ficklin et al., 2016) and
sustained water supplies (Fan et al., 2013). Surface flow, also referred to as fast flow, results from
rapid processes like the saturation or infiltration of excess overland flow and swift subsurface
flow (Beven and Kirkby, 1979), leading to immediate water movement. Surface flow occurs more
rapidly and with more drastic changes than baseflow, which is primarily responsible for flood
generation (Yin et al., 2018) and soil erosion (Morgan, 2011).
Most current studies focus on total runoff variability and attribution, and the relevant
researches are fairly mature (Berghuijs et al., 2017; Han et al., 2023; Liu et al., 2021). However,
few studies pay attention to comprehensive research on the different runoff components (Li et al.,
2020; Liu et al., 2019), and the attributions of $Q_s$ and $Q_b$ changes are still unclear (Hellwig and
Stahl, 2018). Baseflow and surface flow represent different hydrological processes, and their
implications for watershed management are also not identical (Zheng Mingguo, 2014). For





example, the research conducted by Ficklin et al. (2016) in the United States points out apparent
spatial differences between $Q_b$ and $Q_s$ in different seasons. Therefore, it is necessary to quantify
runoff components and distinguish their controlling factors to better understand the runoff
dynamics and facilitate water resources management in the context of intensified climate change
and anthropogenic disturbance.
Unlike $Q$, which is ascertainable through direct observation at hydrological gauges, $Q_b$ and $Q_s$
can only be estimated through indirect methods, including baseflow separation (Wu et al., 2019;
Zhang et al., 2017), isotope tracing (Hale et al., 2022; Wallace et al., 2021) and hydrological
modeling (Al-Ghobari et al., 2020; Cheng et al., 2020; Huang et al., 2007; Kaleris and Langousis,
2017). The first two methods estimate $Q_b$ initially, and $Q_s$ is then derived as the difference
between the $Q$ and the estimated $Q_b$, limiting their ability to examine the dynamic variations of
each runoff component independently, and the isotope tracing method is challenging to conduct
on a large and long-term scale. The hydrological modeling enables to simulate $Q_b$ and $Q_s$
separately, typically reflected in different modules and empirical formulations. In hydrological
models, $Q_b$ is encoded using linear or non-linear storage-discharge functions (Chen and Ruan,
2023; Cheng et al., 2020). $Q_s$ is closely related to rainfall, but the models for estimating it are
usually event-based (such as the Soil Conservation Service Curve Number method (Al-Ghobari et
al., 2020; SCS, 1972; Shi et al., 2017) and very few studies explored the controls on the mean
annual $Q_s$ (Neto et al., 2020). Among various models, the Budyko framework (Budyko, 1974) in
conjunction with water-energy balance method (Choudhury, 1999; Yang et al., 2008) (see the
second row in Table 1), has been widely used in the analysis of mean annual $Q$ due to its simple,
universal and transparent characteristics (He et al., 2022; Roderick and Farquhar, 2011).
Recently, utilizing the extended Budyko framework to estimate $Q_b$ and $Q_s$ has attracted
attention. Wang and Wu (2013) and Neto et al. (2020) established the regression relationship
between baseflow fraction ($BFC$, the ratio of $Q_b$ to precipitation ($P$)) and aridity index ($\phi$, the
ratio of mean annual potential evapotranspiration ($E_0$) to $P$) using analytical formulation.
However, Gnann et al. (2019) reported that using only the $\phi$ struggles to delineate baseflow
variability in humid catchments, where the impact of soil water storage capacity ($S_p$) is as critical
as that of the $\phi$. Thus, Cheng et al. (2021) proposed an analytical curve for describing mean
annual $Q_b$ by introducing $S_p$ as another theoretical boundary. Results show that the developed





curve agrees well with the observed $BFC$ ($R^2 = 0.75$, RMSE = 0.058) and $Q_b$ ($R^2 = 0.86$,
RMSE = 0.19 mm), outperforming the original Budyko framework. Analogously, Yao et al.
(2021) derived similar functions incorporated the $\phi$, $S_p$ and a shape parameter to model $BFC$ and
baseflow index ($BFI$, the ratio of $Q_b$ to $Q$). These extended Budyko frameworks accounting for $S_p$
have advantages in simulating $Q_b$. However, $S_p$ is challenging to obtain through observations and
often requires calibration (Cheng et al., 2021) or computation (Yao et al., 2021), adding certain
uncertainties to the model. Notably, the calibration performance of $Q_s$ in equation (1) to obtain $W_p$
(the proxy of $S_p$) in the catchments of China are not always satisfactory, especially in the northern
catchments (Figure S1). Moreover, the complicated forms can bring inherent uncertainties and
these studies have not validated the formulations of $Q_s$, which are derived by subtracting $Q_b$ from
$Q$ or fitting curves (Cheng et al., 2021; Neto et al., 2020), implying that they may overlook the
physical processes represented by surface flow. In the subsequent discussion, the Budyko
framework and extended Budyko equations are collectively referred to as the "Budyko-type
formulations" (Table 1).

Many researchers have observed similar behavior of $Q_b$ to $Q$ (Cheng et al., 2021; Gnann et al.,

2019; Wang and Wu, 2013). Is there a similar behavior for $Q_s$? In a two-stage partitioning theory
(L'vovich, 1979), runoff components are delineated based on the available water at each stage.
Therefore, is there a general framework to unify different runoff components? Although various
functional forms have been proposed for estimating runoff components in the literature, a
universal method that reveals the mechanisms of mean annual runoff components generation and
subsequent quantification and attribution is still in need.

**Table 1**. The Budyko-type formulations for estimating mean annual runoff components

| References | Formulations | Parameters |
|---|---|---|
| Choudhury (1999); Yang et al. (2008) | $Q = P - \dfrac{P \times E_0}{(P^n + E_0^n)^{1/n}}$ | $n$ |
| Wang and Wu (2013) | $\dfrac{Q_b}{P} = 1 - \left[1 + \left(\dfrac{E_0}{P}\right)^{-v}\right]^{-1/v}$ | $v$ |
| Neto et al. (2020) | $f_S(\phi) = \exp(-\phi^a + \delta_S)^b$ <br> $f_B(\phi) = \exp(-\phi^c + \delta_B)^d$ | $a, b, c, d$ <br> $\delta_S = \ln\left(\left[\dfrac{\bar{Q}_S}{\bar{P}}\right]_{max}\right)^{1/b}$ |





$$\delta_{\text{B}} = \ln\left(1 - \left[\frac{\bar{Q}_{\text{S}}}{\bar{P}}\right]_{max}\right)^{1/d}$$

Cheng et al. (2021)

$$\frac{Q_s}{P} = -\frac{E_0 + S_p}{P} + \left[1 + \left(\frac{E_0 + S_p}{P}\right)^{\alpha_1}\right]^{1/\alpha_1}$$

$$\frac{Q_b}{P} = \frac{S_p}{P} + \left[1 + \left(\frac{E_0}{P}\right)^{\alpha_2}\right]^{1/\alpha_2}$$

$$- \left[1 + \left(\frac{E_0 + S_p}{P}\right)^{\alpha_2}\right]^{1/\alpha_2}$$

$S_p,\ \alpha_1,\ \alpha_2$

$Q_b$

Yao et al. (2021)

$$= \frac{P + S_b - \sqrt{(P + S_b)^2 - 2aS_bP}}{a}\left[1\right.$$

$$- \frac{1 + \frac{E_0}{P}\frac{P}{S_b} - \sqrt{\left(1 + \frac{E_0}{P}\frac{P}{S_b}\right)^2 - 2a\frac{E_0}{P}\frac{P}{S_b}}}{a}\left.\right]$$

$$Q = P - \frac{\frac{P}{S_b} + 1 - \sqrt{\left(\frac{P}{S_b} + 1\right)^2 - 2a\frac{P}{S_b}}}{a}$$

$$* \frac{E_0 + S_b - \sqrt{(E_0 + S_b)^2 - 2aE_0S_b}}{a}$$

$S_b$, a

---

Note that $P$ is the mean annual precipitation, $E_0$ is the mean annual potential evapotranspiration, $f_{\text{S}}(\phi)$ and
$f_{\text{B}}(\phi)$ are the surface flow and baseflow function, respectively and $S_{\text{p}}$ is the catchment storage capacity.
To address these questions, we derived a modified two-stage partitioning framework through
observational data analysis and theoretical derivation based on the Ponce-Shetty model (Ponce
and Shetty, 1995; Sivapalan et al., 2011) (namely the MPS model) at mean annual time scale. The
Ponce-Shetty model is a conceptual model with physical constraint developed at annual scale to
depict how precipitation is partitioned, stored and released in the catchment (Gnann et al., 2019).
It posits that annual precipitation is partitioned into $Q_{\text{s}}$ and soil wetting ($W$) and, subsequently, the
resulting $W$ is partitioned into $Q_{\text{b}}$ and vaporization ($V$) (Sivapalan et al., 2011). The MPS model





enables large-sample catchments research, which may lead to new understanding of mean annual
water balance and allocation.
In general, the objectives of this study are to (1) develop a general and concise formulation to
describe runoff components variability at mean annual time scale; (2) validate and compare the
performance of the developed formulation against Budyko-type formulations; (3) attribute the
variations of runoff components induced by the changes of precipitation and other factors. Here,
we modify the Ponce-Shetty model according to some conditions and hypothesize a general
runoff components model (the MPS model), that describes $Q_s$, $Q_b$ and $Q$ as a function of
respective available water with one parameter. The MPS model is then validated over 662
catchments across China and the contiguous United States (the CONUS) over a wide range of
hydro-meteorological circumstances. The performance of the MPS model is also compared with
the Budyko-type formulations. Section 2 introduces the derivation of the MPS model. Section 3
provides the study catchments, data and the parameter estimation technique. Section 4 shows the
results followed by a discussion in Section 5. The conclusions are summarized in Section 6.

## 2. Derivation of the Modified Ponce-Shetty Model

L'vovich (1979) proposed a conceptual theory for the two-stage catchment water balance
partition at the annual time scale according to Horton's approach (Horton, 1933). Firstly,
precipitation is partitioned into surface flow ($Q_s$) and catchment wetting ($W$, stored water), and
then, the catchment wetting is partitioned into baseflow ($Q_b$) and vaporization ($V$, including
interception loss, evaporation and transpiration). Ponce and Shetty (1995) conceptualized the
partition of each step as the form of a competition, and derived the formulations of runoff
components based on the proportionality hypothesis. Sivapalan et al. (2011) reintroduced the
Ponce-Shetty equations as follows:
In the first stage, $P = Q_s + W$:

$$Q_s = \begin{cases} 0, & if\ P \leq \lambda_s W_p \\ \dfrac{(P - \lambda_s W_p)^2}{P + (1 - 2\lambda_s)W_p}, & if\ P > \lambda_s W_p \end{cases} \tag{1}$$



$$W = \begin{cases} P, & if\ P \leq \lambda_s W_p \\ P - \dfrac{(P - \lambda_s W_p)^2}{P + (1 - 2\lambda_s)W_p}, & if\ P > \lambda_s W_p \end{cases} \tag{2}$$

$$P \to \infty, Q_s \to P - W_p, W \to W_p \tag{3}$$

In the second stage, $W = Q_b + V$:

$$Q_b = \begin{cases} 0, & if\ W \leq \lambda_b V_p \\ \dfrac{(W - \lambda_b V_p)^2}{W + (1 - 2\lambda_b)V_p}, & if\ W > \lambda_b V_p \end{cases} \tag{4}$$

$$V = \begin{cases} W, & if\ W \leq \lambda_b V_p \\ W - \dfrac{(W - \lambda_b V_p)^2}{W + (1 - 2\lambda_b)V_p}, & if\ W > \lambda_b V_p \end{cases} \tag{5}$$

$$W \to \infty, Q_b \to W - V_p, V \to V_p \tag{6}$$

where $\lambda_s$ and $\lambda_b$ are the surface flow and baseflow initial abstraction coefficients, respectively,
which range from 0 to 1. The larger value of $\lambda$, the more difficult it is to generate flow. $W_p$ and $V_p$
are catchment wetting potential and vaporization potential, respectively, which are greater than 0.
The relative $\lambda_s W_p$ and $\lambda_b V_p$ are the surface flow and baseflow generation thresholds,
respectively.
Note that the interannual water storage change is supposed to be negligible (Ponce and Shetty,
1995). In a companion paper of Sivapalan et al. (2011), Harman et al. (2011) employed the
annual Ponce-Shetty model at mean annual time scale and validated its applicability. Using the
first phase as an example, $Q_s$ can be considered a function of $\lambda_s$, denoted as $f(\lambda_s)$:

$$f(\lambda_s) = \begin{cases} 0, & if\ \lambda_s \geq P/W_p \\ \dfrac{(P - \lambda_s W_p)^2}{P + (1 - 2\lambda_s)W_p}, & if\ \lambda_s < P/W_p \end{cases} \tag{7}$$

When $\lambda_s < P/W_p$, the Taylor expansion of $f(\lambda_s)$ at $\lambda_s = 0$ is:

$$f(\lambda_s) = f(0) + f'(0) * \lambda_s + \frac{f''(0)}{2!} * \lambda_s^2 + \cdots + \frac{f^n(0)}{n!} * \lambda_s^n + \cdots \tag{8}$$

Hence, we have the zeroth-order approximation:

$$f(\lambda_s) \approx \frac{P^2}{P + W_p} \tag{9}$$





151 When the remainder term is relatively small, an approximation equation can be used to

152 estimate the multi-year $Q_s$ as:

$$Q_s = \frac{P^2}{P + W_p} \tag{10}$$

153 In addition, the zeroth-order approximation of $Q_b$ can be similarly obtained as:

$$Q_b = \frac{W^2}{W + V_p} \tag{11}$$

154 To evaluate the impact of the remainder term, we calculate the relative bias ($\delta$) of runoff

155 components for 312 basins in China and 350 basins in the United States using the approximate

156 equations (Eq (10) and Eq (11)) and the original Ponce-Shetty equations (Eq (1) and Eq (4)) (data

157 sources in Section 3.1). The parameters in the original Ponce-Shetty equations are calibrated

158 using the nonlinear least squares method. The $\delta$ is calculated as:

$$\delta = \frac{\left| \widetilde{Q_y} - Q_y \right|}{Q_y} \tag{12}$$

159 where $Q_y$ represents runoff components estimated by the Ponce-Shetty equations, and $\widetilde{Q_y}$

160 represents runoff components estimated by the sapproximate equations (Eq (10) and Eq (11)).

161 The spatial distribution of $\delta$ and the cumulative distribution functions (CDFs) of $\delta$ are

162 shown in Figure 1 and Figure 2, respectively. As shown in Figure 1, 77% of the basins have an $\delta$

163 of less than 5%. The average $\delta$ for estimating $Q_s$ is 6.5% in China and 4.8% in the United States,

164 while the average $\delta$ for estimating $Q_b$ is 7.9% in China and 6.6% in the United States, with

165 larger deviations observed in arid basins. Figure 2 indicate that the $\delta$ values for the approximate

166 model are within acceptable limits across both China and CONUS. The relatively low 95%

167 threshold values, particularly for the USA datasets, suggest that the majority of predictions fall

168 within a narrow error range, indicating robust model performance. This acceptability of $\delta$ across

169 regions and variables highlights the approximate equations' capability to maintain prediction

170 accuracy under varying geographical and hydrological conditions, indicating that the Zeroth-order

171 approximation is representative for the original Ponce-Shetty model.



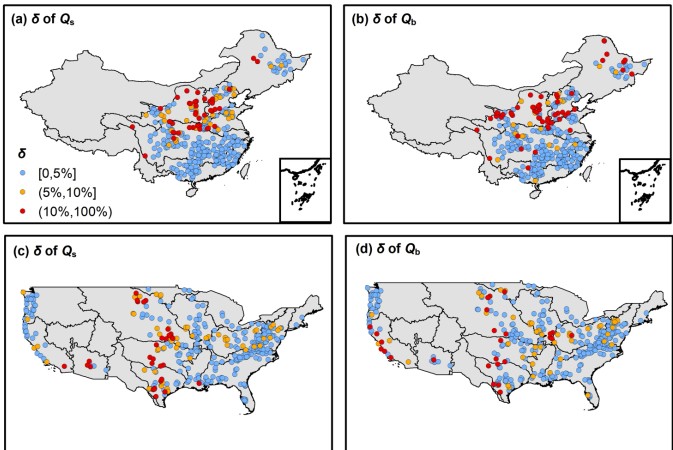

**Figure 1**. The distribution of relative bias ($\delta$) between the results by the approximate equations (Eq (10) and Eq (11)) versus the original Ponce-Shetty equations (Eq (1) and Eq (4)). The first row shows the results for 312 basins in China, and the second row shows the results for 350 basins in CONUS. The first column corresponds to surface flow ($Q_s$), and the second column corresponds to baseflow ($Q_b$).

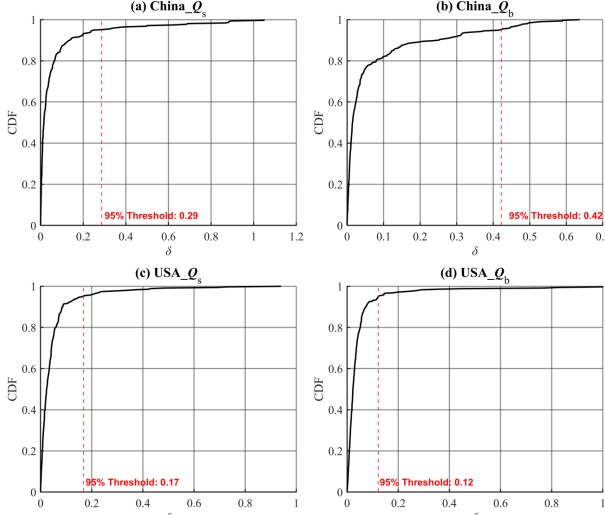

178

**Figure 2**. Cumulative distribution functions (CDFs) of the relative bias ($\delta$) for each dataset, represented by four subplots corresponding to different regions and variables: (a) China_$Q_s$, (b) China_$Q_b$, (c) USA_$Q_s$, and (d) USA_$Q_b$. Each subplot includes a red dashed line indicating the 95% $\delta$ threshold





Therefore, we can approximately consider that on a multi-year scale, $Q_s$ and $Q_b$ can be
estimated using the zeroth-order approximation in Eq (10) and Eq (11). We subsequently assume
a similar formulation of mean annual $Q$:

$$Q = \frac{P^2}{P + U_p} \tag{13}$$

where $U_p$ is the parameter representing the upper limit of the portion remaining after precipitation
is allocated to runoff, hereafter we refer to $U_p$ as evapotranspiration potential.
Integrating equations (10), (11) and (13), we conclude a general formulation to depict
multi-year variability of runoff components and their quantification, hereafter referred to as the
modified Ponce-Shetty model (the MPS model):

$$Q_y = \frac{X^2}{X + M} \tag{14}$$

where $Q_y$ represents runoff components (i.e., $Q$, $Q_s$, $Q_b$), $X$ corresponds to the available water of
each runoff component, i.e., $P$ is the available water of $Q$ and $Q_s$, and $W$ the available water of $Q_b$.
$M$ is an integrated parameter, representing the comprehensive effects of catchment characteristics
and atmospheric water and energy demand.
The MPS model encodes runoff components as a function of available water with only one
parameter, which not only considers processes of runoff generation with physical constraints, but
also, compared to the Budyko-type formulations and the original Ponce-Shetty model, is more
concise in form and requires fewer parameters. Therefore, it is possible to estimate the long-term
runoff components when only long-term variables are known.

## 3. Data and Methodology

### 3.1. Data

To validate the reliability of the MPS model, daily hydrological and meteorological data from
312 catchments in China (Li et al., 2024) and 350 catchments in the CONUS are collected. The
location of all the catchments hydrological stations is shown in Figure 3.





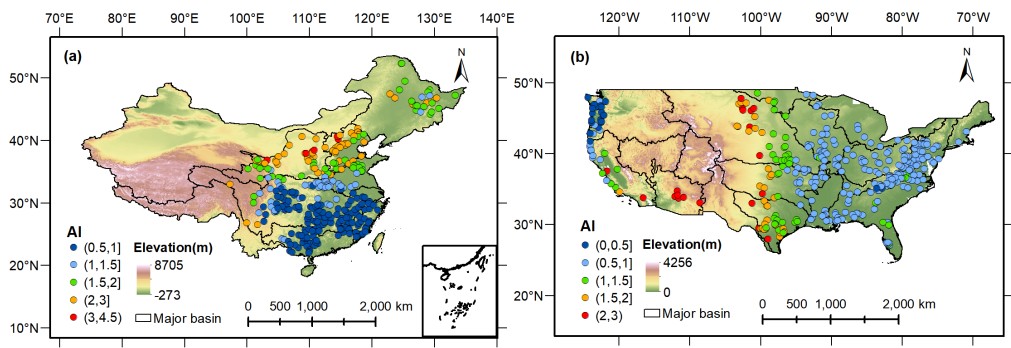


**Figure 3**. Location of hydrological stations for the (a) 312 catchments in China and (b) 350

catchments in the CONUS, colored by the value of aridity index ($\phi$, namely $E_0/P$).

In China, precipitation data at $0.25°$ spatial resolution are obtained from the China

Gauge-based Daily Precipitation Analysis (CGDPA) (Shen and Xiong, 2016). Other

meteorological data, including wind speed, sunshine hours, relative humidity, and air temperature,

are from about 736 stations of the China Meteorological Data Service Center

(http://data.cma.cn/en, last access: 11 November 2023). The in-site meteorological data are

interpolated into a 10-km grid using the inverse-distance weighted method (Yang et al., 2014).

We use the Penman equation (Penman, 1948) to estimate $E_0$ of each grid. The aridity index $\phi$ is

subsequently calculated as $E_0/P$. All grid data are aggregated and lumped for individual

catchments. The discharge data are collected from the Hydrological Bureau of the Ministry of

Water Resources of China (https://www.mwr.gov.cn/english/, last access: 20 December 2023)

and are selected based on the length of records exceeding 35 years with less than 5% missing data.

The time range for all data is 1960-2000.

In the CONUS, we use data set from CAMELS (Addor et al., 2017; Newman et al., 2015). The

CAMELS data set provides 662 catchments with daily time series of precipitation and observed

runoff along with aridity index, and most catchments contain 35 years of continuous runoff from

1980 to 2014. The criteria for excluding catchments are referred to Gnann et al. (2019), and

finally 350 catchments remained.

We use the one-parameter Lyne-Hollick digital filter (Lyne, 1979) to separate daily $Q_s$ and $Q_b$

from daily $Q$. The Lyne-Hollick method is applied forward, backward, and forward again with a

filter parameter of 0.925 and has manifested to be reliable to obtain runoff components (Lee and





Ajami, 2023). We use the separated $Q_s$ and $Q_b$ as the reference. Although there are other baseflow
separation algorithms, according to Troch et al. (2009), the choice of baseflow separation
algorithm is not a significant determinant of the water balance at the annual scale.
All the hydrological and meteorological data are aggregated to the annual and mean annual
time scales for further analysis.
**3.2. Calibration and Validation**
Spatially, to verify the MPS model's ability to characterize the variability of runoff components
between catchments, we utilize the least squares fitting algorithm to estimate parameters, i.e., $W_p$,
$V_p$ and $U_p$. The three parameters are restricted to being between 0 mm and 50, 000 mm, which is
considered high enough to not affect the parameter estimation (Gnann et al., 2019).
In terms of time, we split all data into two periods for parameter calibration and validation of
Eq. (14) for individual catchments. In China, the data ranges from 1960 to 2000, so we use the
first 31 years (1960-1990) as the calibration period and the remaining 5-10 years (1991-2000) as
the validation period. In the CONUS, the calibration period is chosen as 1980-2000, and the
validation period is from 2001 to 2014. When we know mean annual $Q_s$, $Q_b$, $Q$, $P$ and $W$ of the
first period, the parameters, i.e., $W_p$, $V_p$ and $U_p$, can be derived from Eq. (14). Postulating the
parameters remain unchanged during two periods, we consequently can estimate the mean annual
$Q_s$, $Q_b$ and $Q$ of the second period using Eq. (14). Note that the catchment wetting $W$ is calculated
as the difference of the $P$ and estimated $Q_s$.
The surface flow fraction (*SFC*, the ratio of surface flow to precipitation) and baseflow fraction
(*BFC*, the ratio between baseflow and precipitation) represent the proportion of rainfall becoming
different runoff components, which are commonly used to quantity surface flow and baseflow
(Wang and Wu, 2013). Therefore, we evaluate the simulation of *SFC* and *BFC* as well as the
volume of runoff components.
The performance of the MPS model is evaluated by the coefficient of determination ($R^2$) and
the root mean square error (RMSE):

$$R^2 = \left( \frac{\sum_{i=1}^{N}\left(X_{sim,i} - \bar{X}_{sim}\right)\left(X_{obs,i} - \bar{X}_{obs}\right)}{\sqrt{\sum_{i=1}^{N}\left(X_{sim,i} - \bar{X}_{sim}\right)^2 \sum_{i=1}^{N}\left(X_{obs,i} - \bar{X}_{obs}\right)^2}} \right)^2 \tag{15}$$





$$RMSE = \sqrt{\frac{1}{N}\sum_{i=1}^{N}\left(X_{sim,i} - X_{obs,i}\right)^2} \tag{16}$$

where $X$ represents the evaluated variable, i.e., mean annual $Q$, $Q_s$ and $Q_b$, $SFC$ and $BFC$ in this
study. The subscript $obs$ and $sim$ represents the observed and simulated value, respectively.
Higher $R^2$ and lower RMSE indicate good model performance.

### 3.3. Attribution analysis

We split the data into the first period (1960-1990 in China and 1980-2000 in the CONUS) and
the second period (1991-2000 in China and 2001-2014 in the CONUS) to attribute runoff
components variation between two periods. Note that the attribution of $\Delta Q$ is only conducted in
China because the $E_0$ in CAMELS dataset is a constant in each catchment. In the MPS model, we
consider that the runoff changes between two long-term periods are caused by available water and
other environmental and anthropogenic factors (such as land cover/use change and
evapotranspiration variation) encoded by parameters. For the changes of surface flow ($\Delta Q_s$) and
total runoff ($\Delta Q$), postulating that each variable is independent in the MPS model, the first-order
approximation of the $\Delta Q_s$ and $\Delta Q$ from the second period to the first period can be expressed as
(Milly and Dunne, 2002):

$$\Delta Q_s = \frac{\partial Q_s}{\partial P}\Delta P + \frac{\partial Q_s}{\partial W_p}\Delta W_p \tag{17a}$$

$$\Delta Q = \frac{\partial Q}{\partial P}\Delta P + \frac{\partial Q}{\partial U_p}\Delta U_p \tag{17b}$$

where the two terms on the right side of equation (17a) respectively represent changes in $Q_s$
caused by changes in $P$ ($\Delta Q_{s-P}$) and other factors ($\Delta Q_{s-Wp}$), and the two terms on the right side
of equation (17b) respectively represent changes in $Q$ caused by changes in $P$ ($\Delta Q_P$) and other
factors ($\Delta Q_{Wp}$). For convenience, we refer partial derivative coefficient $\frac{\partial Q_s}{\partial P}$, $\frac{\partial Q_s}{\partial W_p}$, $\frac{\partial Q}{\partial P}$ and $\frac{\partial Q}{\partial U_p}$ in
equation (17) as $\zeta_{Qs-P}$, $\zeta_{Qs-Wp}$, $\zeta_{Q-P}$ and $\zeta_{Q-Wp}$, which can be calculated as:

$$\zeta_{Qs-P} = \frac{P^2 + 2PW_p}{\left(P + W_p\right)^2} \tag{18a}$$





$$\zeta_{Qs-Wp} = \frac{-P^2}{\left(P + W_{\mathrm{p}}\right)^2} \tag{18b}$$

$$\zeta_{Q-P} = \frac{P^2 + 2PU_{\mathrm{p}}}{\left(P + U_{\mathrm{p}}\right)^2} \tag{18c}$$

$$\zeta_{Q-Wp} = \frac{-P^2}{\left(P + U_{\mathrm{p}}\right)^2} \tag{18d}$$

The changes of baseflow ($\Delta Q_{\mathrm{b}}$) is induced by the variations of the $W$ and $V_{\mathrm{p}}$. However, we
focus more on the impact of $P$ in application. Therefore, we combine equation (10), (11) and $W =$
$P$-$Q_{\mathrm{s}}$, so the $Q_{\mathrm{b}}$ can be calculated as :

$$Q_{\mathrm{b}} = \frac{P^2 W_{\mathrm{p}}^2}{\left(P + W_{\mathrm{p}}\right)\left(PW_{\mathrm{p}} + PV_{\mathrm{p}} + W_{\mathrm{p}}V_{\mathrm{p}}\right)} \tag{19}$$

The $\Delta Q_{\mathrm{b}}$ can be attributed as the variations of $P$, $W_{\mathrm{p}}$ and $V_{\mathrm{p}}$:

$$\Delta Q_{\mathrm{b}} = \frac{\partial Q_{\mathrm{b}}}{\partial P}\Delta P + \frac{\partial Q_{\mathrm{b}}}{\partial W_{\mathrm{p}}}\Delta W_{\mathrm{p}} + \frac{\partial Q_{\mathrm{b}}}{\partial V_{\mathrm{p}}}\Delta V_{\mathrm{p}} \tag{20}$$

where the three terms on the right side of equation (20) respectively represent changes in $Q_{\mathrm{b}}$
caused by changes in $P$ ($\Delta Q_{\mathrm{b}-P}$), $W_{\mathrm{p}}$ ($\Delta Q_{\mathrm{b}-\mathrm{Wp}}$) and $V_{\mathrm{p}}$ ($\Delta Q_{\mathrm{b}-\mathrm{Vp}}$). The partial derivative
coefficient $\frac{\partial Q_{\mathrm{b}}}{\partial P}$ ($\zeta_{Qb-P}$), $\frac{\partial Q_{\mathrm{b}}}{\partial W_{\mathrm{p}}}$ ($\zeta_{Qb-Wp}$) and $\frac{\partial Q_{\mathrm{b}}}{\partial V_{\mathrm{p}}}$ ($\zeta_{Qb-Vp}$) can be calculated as:

$$\zeta_{Qb-P} = \frac{2P^2 W_{\mathrm{P}}^3 V_{\mathrm{p}} + P^2 W_{\mathrm{P}}^4 + 2PW_{\mathrm{P}}^4 V_{\mathrm{p}}}{\left(P + W_{\mathrm{p}}\right)^2 \left(PW_{\mathrm{p}} + PV_{\mathrm{p}} + W_{\mathrm{p}}V_{\mathrm{p}}\right)^2} \tag{21a}$$

$$\zeta_{Qb-Wp} = \frac{P^4 W_{\mathrm{P}}^2 + 2P^4 W_{\mathrm{p}}V_{\mathrm{p}} + 2P^3 W_{\mathrm{P}}^2 V_{\mathrm{p}}}{\left(P + W_{\mathrm{p}}\right)^2 \left(PW_{\mathrm{p}} + PV_{\mathrm{p}} + W_{\mathrm{p}}V_{\mathrm{p}}\right)^2} \tag{21b}$$

$$\zeta_{Qb-Vp} = \frac{-P^2 W_{\mathrm{P}}^2}{\left(P + W_{\mathrm{p}}\right)^2 \left(PW_{\mathrm{p}} + PV_{\mathrm{p}} + W_{\mathrm{p}}V_{\mathrm{p}}\right)^2} \tag{21c}$$

To verify the applicability of the MPS model for runoff components attribution, we compare
the calculated $\Delta Q_{\mathrm{s}}$, $\Delta Q_{\mathrm{b}}$ and $\Delta Q$ using equation (17) and (20) with the observed $\Delta Q_{\mathrm{s}}$, $\Delta Q_{\mathrm{b}}$
and $\Delta Q$ between two periods. The evaluation metrics are $R^2$ and RMSE.
The relative contribution ratios of $P$ and other factors to runoff components change are
calculated as:





$$\eta_P = \frac{\Delta Q_{y-P}}{|\Delta Q_{y-P}| + |\Delta Q_{y-W\mathrm{p}}| + |\Delta Q_{y-V\mathrm{p}}|} \times 100\% \tag{22a}$$

$$\eta_{W\mathrm{p}} = \frac{\Delta Q_{y-W\mathrm{p}}}{|\Delta Q_{y-P}| + |\Delta Q_{y-W\mathrm{p}}| + |\Delta Q_{y-V\mathrm{p}}|} \times 100\% \tag{22b}$$

$$\eta_{V\mathrm{p}} = \frac{\Delta Q_{y-V\mathrm{p}}}{|\Delta Q_{y-P}| + |\Delta Q_{y-W\mathrm{p}}| + |\Delta Q_{y-V\mathrm{p}}|} \times 100\% \tag{22c}$$

where $\eta_P$, $\eta_{W\mathrm{p}}$ and $\eta_{V\mathrm{p}}$ are the relative contribution ratios of $P$, $W_\mathrm{p}$ and $V_\mathrm{p}$ to runoff
components, respectively. We subsequently use the absolute values of $\eta$ to identify the dominant
factor impacting runoff components.
**4. Results**
**4.1. Inter-Catchment Variability of Runoff Components**
We employ the MPS model to fit the relationship between mean annual available water and
runoff components. In China, as shown in Figure 4(a-c), the MPS model performs well in
describing runoff components variability between catchments, with $R^2$ values of 0.86, 0.68 and
0.91 for fitting $Q_\mathrm{s}$, $Q_\mathrm{b}$ and $Q$, respectively. The solid lines are the best-fitted MPS curves derived
using the least squares fitting algorithm, implying the median values of different parameters. We
also give the potential upper and lower limits of $W_\mathrm{p}$, $V_\mathrm{p}$ and $U_\mathrm{p}$ across catchments. Similarly,
Figure 4(d-f) illustrates that the MPS model achieves good fitting in the CONUS, with $R^2$ of 0.81,
0.44 and 0.80 for fitting $Q_\mathrm{s}$, $Q_\mathrm{b}$ and $Q$, respectively. The fitted parameters in the CONUS are
smaller than those in China, while they have more comprehensive ranges between catchments,
meaning a more significant heterogeneity in climate and underlying surface.
Figures 4 demonstrates that the MPS model can effectively reproduce the spatial variability of
different runoff components along with the aridity index ($E_0/P$), which are primarily controlled by
the available water of the corresponding partition stage. The performance of MPS model to fit $Q_\mathrm{s}$
and $Q$ is better than that of $Q_\mathrm{b}$, indicating that the factors controlling $Q_\mathrm{b}$ are more complicated and
not fully reflected in the model. With catchment properties and other factors (integrated by the
parameters in the MPS model) remaining unchanged, the more the available water, the higher the



runoff generated. Conversely, smaller parameter values are associated with greater runoff for a

given amount of available water.

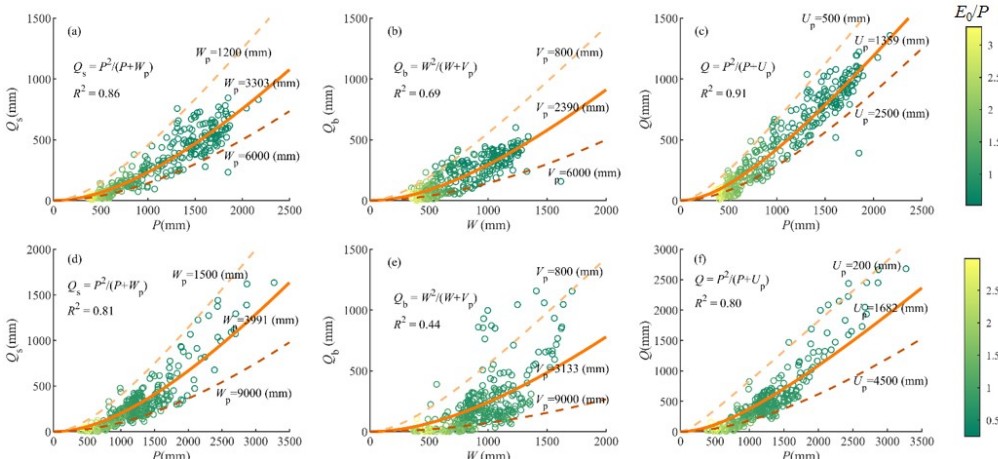

**Figure 4**. The MPS model relating (a) $P$ versus $Q_s$, (b) $W$ versus $Q_b$ and (c) $P$ versus $Q$ in China

and (d) $P$ versus $Q_s$, (e) $W$ versus $Q_b$ and (f) $P$ versus $Q$ in the CONUS. The lines are the fitted

MPS curves with best fitting (solid line) and potential upper limit and lower limit (dashed lines)

parameters.

**4.2. Validation of Runoff Components Estimation**

Figure 5 shows the estimated mean annual $Q_s$, $Q_b$ and $Q$ in validation periods using the MPS

model with inverted parameters in equation (14) in China and the CONUS. The simulated runoff

components match very well with the observed, with $R^2$ greater than 0.97 and RMSE less than 66

mm. There is no significant difference in the performance in simulating $Q_s$, $Q_b$, and $Q$, except for

a slight underestimation in simulating $Q_b$ of catchments in China and some in the CONUS.

In panels (a), (b), and (c), we observe that the scatter points for both China (red circles) and the

CONUS (blue circles) are closely aligned with the 1:1 line, further underscoring the strong

correlation between modeled and observed values. Specifically, the results show that the MPS

model effectively captures surface flow ($Q_s$), baseflow ($Q_b$), and total runoff ($Q$) for both regions.

Despite the generally good performance, a slight underestimation of $Q_b$ is evident in a subset of

catchments in China and, to a lesser extent, in the CONUS. However, these discrepancies are

minimal and do not significantly detract from the model's overall accuracy.



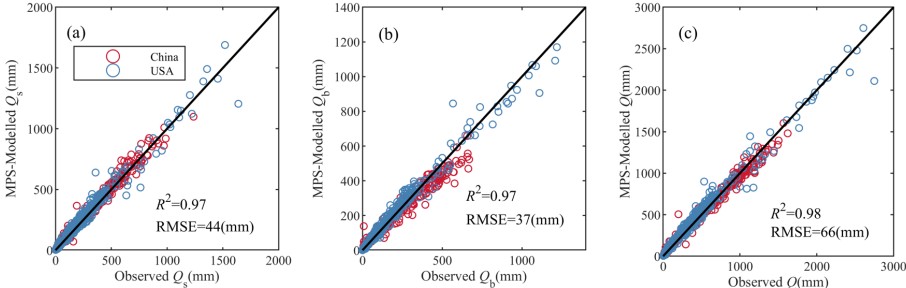


**Figure 5**. The observed and simulated mean annual (a) surface flow, (b) baseflow and (c) total runoff by the MPS model in China (red circles) and the CONUS (blue circles).



Figure 6 presents the estimation of *SFC* and *BFC* in validation periods using the MPS model.
Similar to the simulation of $Q_s$, the two methods also show highly consistent estimation of *SFC*
(panel (a)), with $R^2$ of 0.94 and RMSE of 0.03. This demonstrates the MPS model's robust
capability to estimate the surface flow fraction in China and the CONUS, closely aligning with
the observed data. Panel (b) presents the estimation of *BFC*, where the MPS model achieves
significant accuracy, reflected by the same $R^2$ and RMSE values (0.94 and 0.03, respectively).
This strong performance indicates that the MPS model is highly effective in simulating SFC and
*BFC* across various catchments.

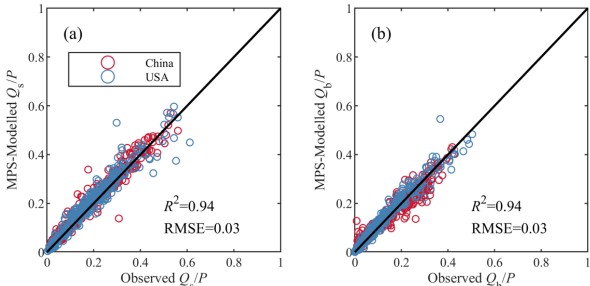


**Figure 6**. The observed and simulated (a) surface flow fraction ($Q_s/P$) and (b) baseflow fraction ($Q_b/P$) by the MPS model in China (red circles) and the CONUS (blue circles).



Figure 5 and Figure 6 document that the MPS model can effectively estimate the multi-year
average of all runoff components and the proportions of precipitation allocated to runoff.
The good validation performance of the MPS model verified our hypothesis that the parameters
in the general formulations remain stable at the mean annual time scale. The parameters reflect



the comprehensive impact of climate and catchment characteristics, i.e., catchment wetting
potential ($W_p$), vaporization potential ($V_p$) and the upper limit of the portion remaining after
precipitation is allocated to runoff ($U_p$). As shown in Figure 7(a-c), the spatial distribution of the
parameters across China exhibits pronounced divergence between the northern and southern
catchments, as well as the eastern and the western. The $W_p$, $V_p$ and $U_p$ exhibit similar spatial
patterns, which can be approximately divided into two tiers from north to south. In the catchments
of the Songliao River Basin in the northeast, the Yangtze River Basin and Pearl River Basins in
the south, the parameters are relatively small, with $W_p$ and $U_p$ ranging from 0 to 2000 mm, and $V_p$
from 0 to 4000 mm, resulting large flow. In the catchments of the Yellow River Basin, Huaihe
River Basin and Haihe River Basin in the north, the parameters are quite large and usually more
than 5000 mm and even 8000 mm, leading to small flow. From west to east, $W_p$ exhibits higher
values in the Yangtze and Yellow Rivers Basin sources, whereas $V_p$ and $U_p$ are smaller in the
source regions. This disparity may reflect variations in the two-stage partition of precipitation,
contributing to spatial differences in total runoff. According to Figure 7(c), we can deduce that
the spatial distribution of higher total runoff in south and lower in north across China, aligning
with previous observational studies (He et al., 2021; He et al., 2022; Yang et al., 2019).

Figure 7(d-f) shows an evident west-east discrepancy of the three parameters across the

CONUS. Typically, $W_p$, $V_p$ and $U_p$ of the catchments in the west coast and eastern regions are less
than 5000 mm, while parameters in the central United States are extensive with values more than
8000 mm. This indicates relatively low flow in the central regions. Notably, the parameters upper
limits in the catchments of the CONUS are significantly higher than those in China. The
extremely large values may be associated with significant parameter uncertainty (Gnann et al.,
2019). Figure 7 demonstrates that the values of the three parameters are larger in arid catchments
and their spatial patterns are similar to that of climate zoning, which provides insights for
parameterization.





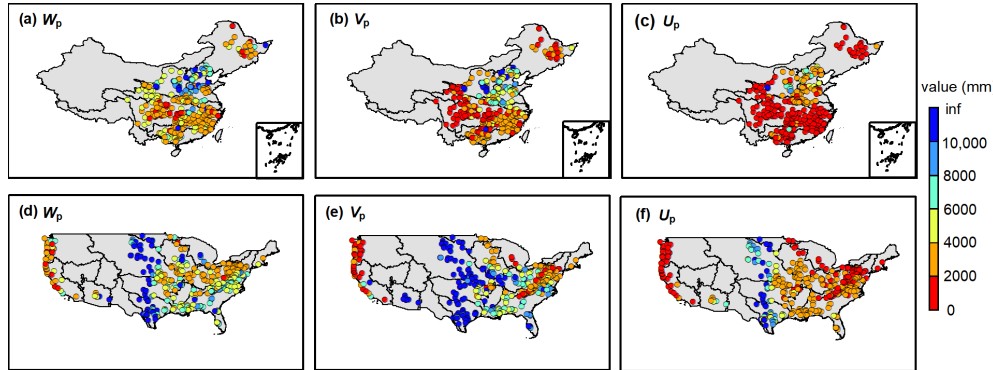


**Figure 7**. The (a) wetting potential ($W_p$), (b) vaporization potential ($V_p$) and (c)

evapotranspiration potential ($U_p$) of the catchments in China and (d) wetting potential ($W_p$), (e)
vaporization potential ($V_p$) and (f) evapotranspiration potential ($U_p$) of the catchments in the

CONUS.

Figure 8 shows the violin plots of the parameters in the catchments of China and the CONUS.
The median values of $W_p$, $V_p$, and $U_p$ in China are 3659 mm, 2220 mm and 1453 mm,
respectively. The median values of $W_p$, $V_p$, and $U_p$ in the CONUS are 4531 mm, 3424 mm and
2385 mm, respectively. Overall parameters in China are smaller and denser than those in the
CONUS, implying a smaller variability of runoff components in China. Furthermore, the $C_v$ value
of $V_p$ (1.6 in China and 6.8 in the CONUS) is the largest, followed by $U_p$ (0.9 in China and 1.6 in
the CONUS), and the smallest for $W_p$ (0.6 in China and 1.5 in the CONUS). This indicates that
the parameter dispersion controlling the second partition stage of rainfall is the greatest, which
could partly account for the challenges in accurately estimating $Q_b$.

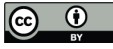



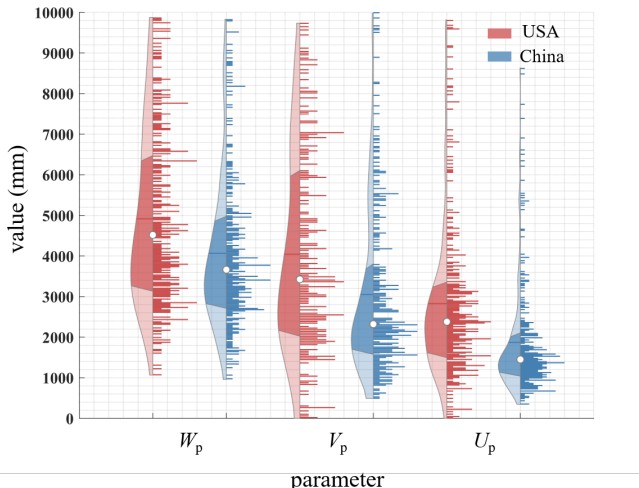


**Figure 8**. Violin plots of the parameters in the catchments of China and the CONUS. In each

violin plot, the left side represents the distribution, with the shaded area indicating the box plot,

the dot representing the mean, and the right side showing the histogram.The length of the

histogram represents the number of catchments (values larger than 10,000 are not shown).

**4.3. The Changes Attribution of Runoff Components**

The metrics to evaluate the attribution results between the changes of the observed and

simulated runoff components are shown in Table 2. We use the MPS model to estimate the

changes of $Q_s$ ($\Delta Q_s$), $Q_b$ ($\Delta Q_b$) and $Q$ ($\Delta Q$) from two long-term periods by equation (17) and

(20), and for comparison, we use the Budyko framework to estimate $\Delta Q$, which is considered as

the changes induced by $P$, $E_0$, and parameter $n$ (the calculation formulations can refer Zhang et al.

(2018)). The estimated and observed runoff components variations exhibit high consistency

(Figure 9), with an $R^2$ of 0.99 and RMSE of 1.6 mm of $\Delta Q_s$ attribution and $R^2$ of 0.88 and RMSE

of 18 mm of $\Delta Q_b$ attribution, respectively. As for $\Delta Q$ attribution, both the MPS model and the

Budyko framework can attain satisfactory performance, while the MPS model has a higher $R^2$

(0.91) than the Budyko framework (0.89). Table 2 demonstrates that the MPS model can

accurately quantify changes in runoff components over two periods. Subsequently, we quantify

the contribution of precipitation and other factors (encoded by parameter $W_p$ and $V_p$) to $\Delta Q_s$ and

$\Delta Q_b$.





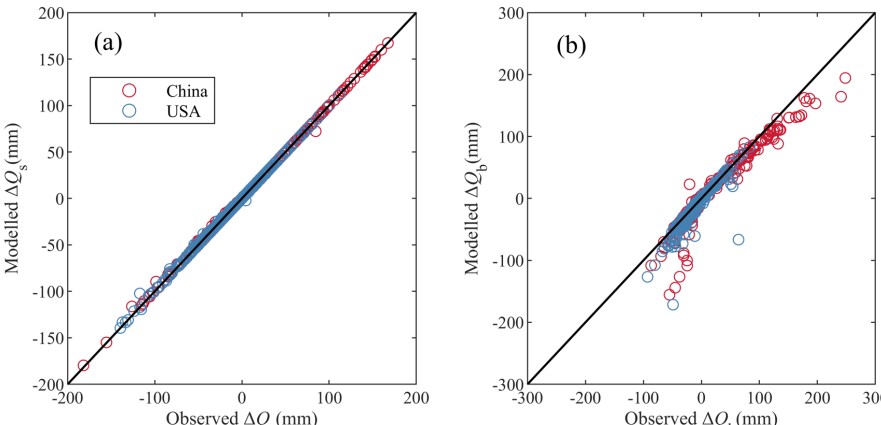

**Figure 9**. The observed and modelled (a) surface flow and (b) baseflow by the MPS model.

**Table 2.** The metrics of the attribution validation

| Variables | $R^2$ | RMSE (mm) |
|---|---|---|
| $\Delta Q_s$ | 0.99 | 1.6 |
| $\Delta Q_b$ | 0.90 | 16 |
| $\Delta Q$ (the MPS model) | 0.91 | 42 |
| $\Delta Q$ (the Budyko framework) | 0.89 | 41 |

Figure 10 shows the $\Delta Q_s$ induced by $P$ ($\Delta Q_{s-P}$) and other factors ($\Delta Q_{s-W_p}$) along with the dominant factor in the catchments of China and the CONUS. From 1960-1990 to 1991-2000 in China, the multi-year variation in $P$ has resulted in $Q_s$ change ranging from -105 to 344 mm, mainly increasing $Q_s$ in the catchments of the Songliao River Basin, the middle and lower Yangtze River Basin, the Southeast River Basin and Pearl River Basin, and decreasing $Q_s$ in the catchments of the Yellow River Basin and the upper Yangtze River Basin (Figure 10(a)). The variations of other factors, such as land use/cover change and human activities, have resulted in $Q_s$ change ranging from -186 to 124 mm, primarily decreases $Q_s$ in 70% catchments (Figure 10(b)). $P$ and other $W_p$ are the dominant factor altering $Q_s$ in southern and northern China, respectively (Figure 10(c)). From 1980-2000 to 2000-2014 in the CONUS, variation in $P$ has resulted in $Q_s$ change ranging from -469 to 149 mm, mainly increasing $Q_s$ in the catchments of Interior Plains (except Great Plains), Coastal Plain, Interior highlands and Appalachian Plain, and decreasing $Q_s$ in the catchments of the Great Plains and Pacific Mountains (the physiographic





divisions are referred to Wu et al. (2021)) (Figure 10(d)). The variations of other factors have
resulted in $Q_s$ change ranging from -230 to 467 mm, primarily decreases $Q_s$ in 75% catchments
(Figure 10(e)). The catchments in the CONUS dominated by $P$ and $W_p$ account for 43% and 57%,
respectively (Figure 10(f)).

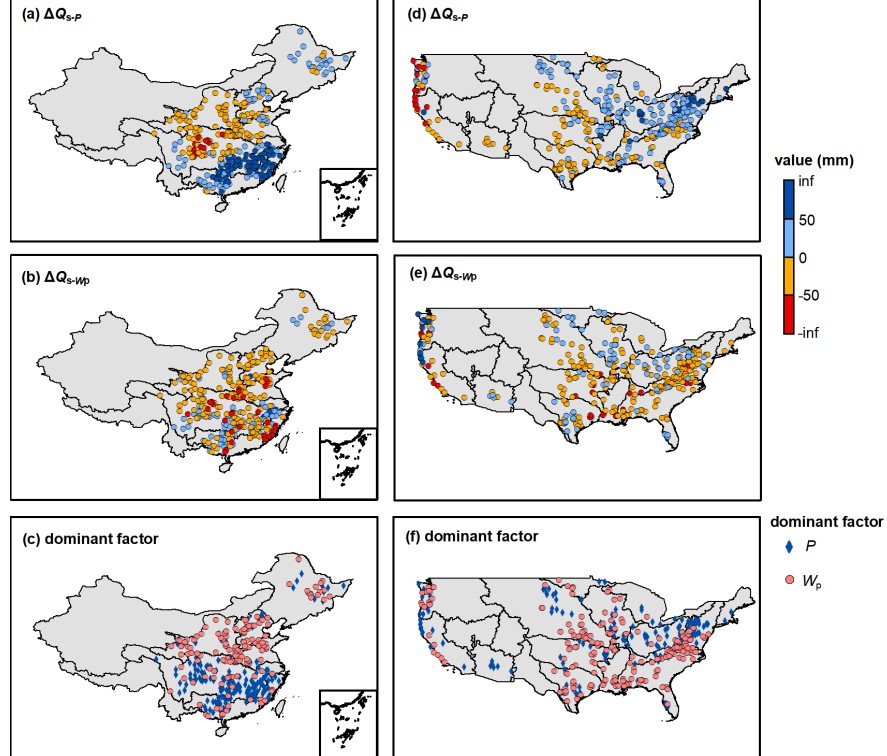


**Figure 10**. The surface flow change induced by precipitation and wetting potential ($W_p$) along
with the dominant controlling factor.
Figure 11 shows the $\Delta Q_b$ induced by $P$ ($\Delta Q_{b-P}$), $W_p$ ($\Delta Q_{b-Wp}$) and $V_p$ ($\Delta Q_{b-Vp}$) in the
catchments of China and the CONUS. The spatial pattern of the effect of $P$ on $Q_b$ is similar to that
of the $Q_s$, resulting in $Q_b$ change from -38 to 79 mm in China (Figure 11(a)) and -129 to 92 mm in
the CONUS (Figure 11(e)), respectively. Catchment wetting potential has a positive effect on $Q_b$
in 70% and 75% catchments of China and the CONUS, respectively (Figure 11(b)and (f)), mainly
in the northern China and the Interior Highlands, Coastal Plain and Appalachian Highlands of the
CONUS. Vaporization potential has a negative effect on $Q_b$ in 56% and 77% catchments of China
and the CONUS, respectively, mainly in the upper Yangze River Basin and northern China and



the central and southeastern CONUS (Figure 11(c)and (g)). Although $V_p$ is the dominant factor
controlling $Q_b$ variation in most catchments in both China (62%) and the CONUS (71%) (Figure
11(d)and (h)), the contributions of the $P$, $W_p$ and $V_p$ are not significantly discrepant in terms of
magnitude.

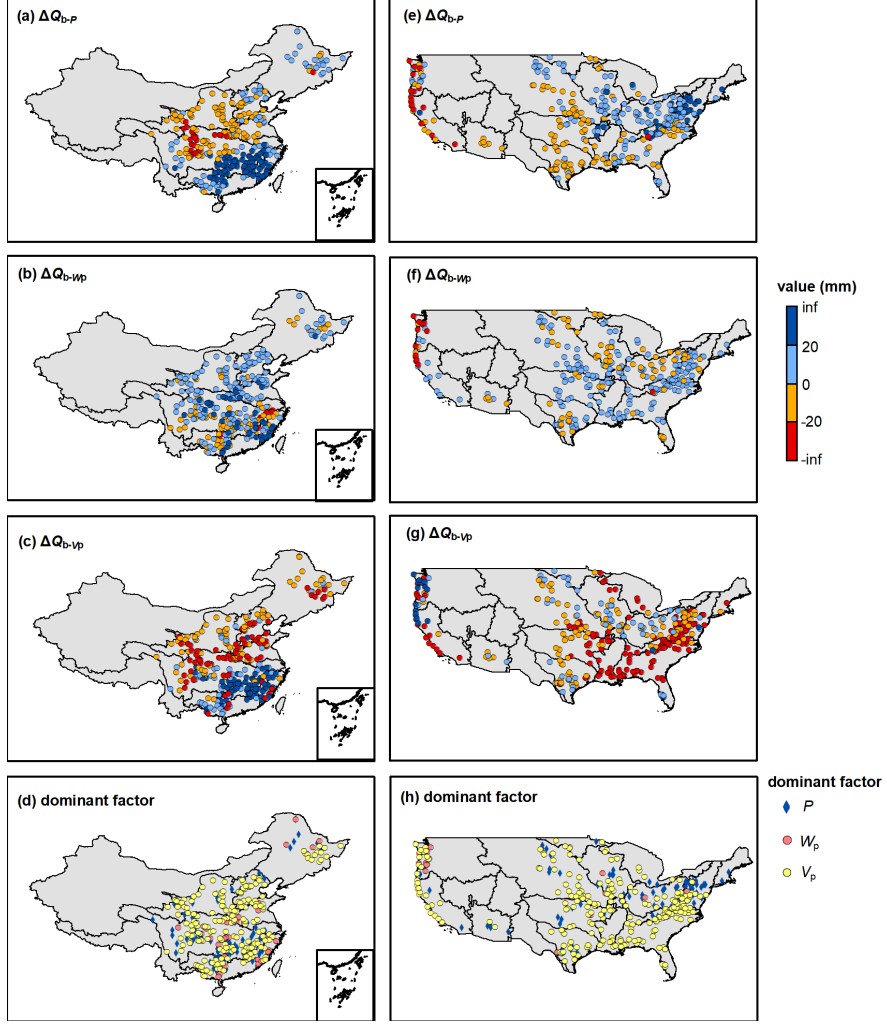


**Figure 11**. The baseflow change induced by precipitation, wetting potential ($W_p$) and
vaporization potential ($V_p$) along with the dominant controlling factor.
Overall, Figure 10 and 11 illustrate that the variation of $Q_s$ is jointly controlled by $P$ and other
factors, while the variation of $Q_b$ is mainly influenced by $V_p$. This demonstrates that $Q_s$ is closely
related to rainfall and soil storage capacity, while $Q_b$ is more affected by catchment attributes,



atmospheric water and energy demand, etc. In regions where runoff components are reduced,
focus should be given to the risks of drought and river discontinuity; conversely, in areas
experiencing runoff components increase, there is a need to guard against the risk of flooding.

## 5. Discussion

### 5.1. Superiorities of the MPS Model

The researches about long-term runoff components quantification and attribution are currently
fragmented and region-specific (Beck et al., 2013; Gnann, 2021). This study has developed a
general formulation (the MPS model) through observational data analysis and theoretical
derivation based on the Ponce-Shetty model, unveiling the patterns of variability in different
runoff components at mean annual time scale. Compared to the commonly used Budyko-type
formulations, it can not only estimate mean annual $Q$ and $Q_b$, but also can depict the variability of
$Q_s$. Figure 12 shows the estimated mean annual runoff components by the Budyko-type
formulations (equations in the second and fifth rows of Table 1 in this paper). The Budyko-type
formulations also achieve good validation performance, with $R^2$ greater than 0.95 and RMSE less
than 78 mm. Although the MPS model and the Budyko-type formulations are comparable in
terms of $R^2$, especially with almost equal simulation results of $Q_s$, the MPS model reduced the
RMSE values by 10 mm and 12 mm for estimating $Q_b$, respectively.

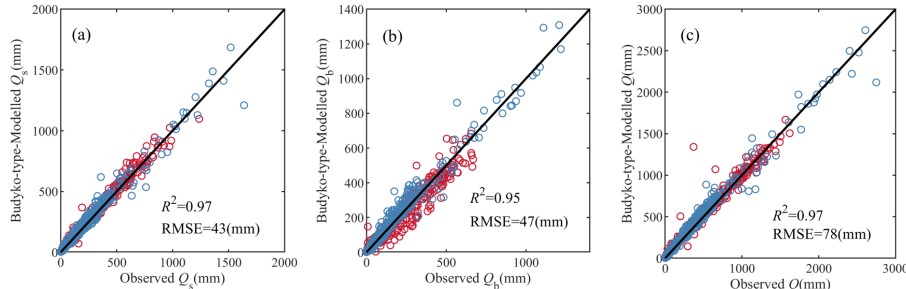

**Figure 12**. The observed and simulated mean annual (a) surface flow, (b) baseflow and (c) total
runoff by the Budyko-type formulations in China (red circles) and the CONUS (blue circles).
Figure 13 presents the estimation of *SFC* and *BFC* in validation periods using the Budyko-type
formulations. The two methods also show highly consistent estimation of *SFC*, with $R^2$ of 0.94





and RMSE of 0.03. However, the Budyko-type formulations underestimate the *BFC* of most
catchments in China, while the MPS model greatly improves the simulation accuracy of *BFC*.

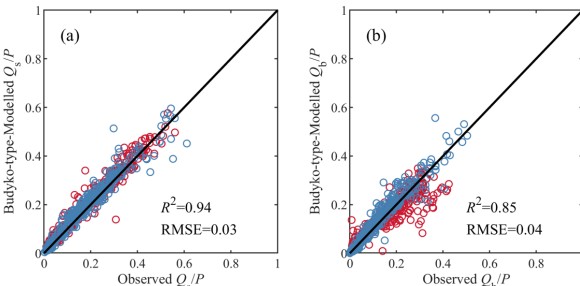

**Figure 13**. The observed and simulated (a) surface flow fraction ($Q_s/P$) and (b) baseflow fraction

($Q_b/P$) by the MPS model in China (red circles) and the CONUS (blue circles).

In conclusion, the MPS model has comparable capability in simulating $Q_s$ and *SFC* to that of
Budyko-type formulations. Moreover, it outperforms Budyko-type formulations in estimating $Q_b$
and $Q$, and reveals superiority in estimating *BFC*. By characterizing runoff components as
functions of available water at corresponding stages with a composite parameter, the MPS model
is more concise in form and eliminates additional and complex parameter computations, thereby
facilitating broader application in large-sample investigations.
In addition to precisely quantifying runoff components and the allocation of precipitation, this
model has innovatively attributed the contributions of different factors on the changes of $Q_s$ and
$Q_b$. Our results show that the variation of $Q_s$ is jointly controlled by $P$ and other factors. $P$ plays
an dominant role in the variation of $Q_s$ in the catchments of the Yangtze River Basin, Southeast
Basin and Pearl River Basin of China and the west coast of the CONUS, where precipitation has
been reported to have undergone significant changes (Li et al., 2021; Mallakpour and Villarini,
2017; Massoud et al., 2020; Xu et al., 2022). This is possibly due to more extreme precipitation
events and summer rainfall in the middle-lower Yangtze River Basin (Ye et al., 2018) and an
increasing trend in the frequency of heavy precipitation over large areas of the CONUS
(Mallakpour and Villarini, 2017). Previous studies reported that the variation of $Q$ in these
regions are dominated by $P$ (He et al., 2022; Huang et al., 2016). Now it seems that $P$ mainly
affects the first allocation stage ($Q_s$) and consequently change total runoff. The variation of $Q_b$ is
mainly influenced by $V_p$, indicating that we should pay more attention to the changes of





catchment attributes, atmospheric water and energy demand in most catchments when
investigating $Q_b$.
Overall, this conceptual model extracted from observed rainfall-runoff data provides a concise,
general and effective tool for predicting runoff components, and evaluating their responses to
climate and environment under global change.

**5.2. Parameter Interpretation**

In the MPS model, each runoff component is associated with a parameter that can be
interpreted as the upper limit of the remaining portion of available water after it has been
partitioned into runoff at each stage. For instance, in the first stage, precipitation is allocated to
surface flow and catchment wetting, with $W_p$ representing the upper limit of catchment wetting,
which describes the catchment's storage capacity related to soil, topography and so on (Cheng et
al., 2023). For the second stage, the available water comes from catchment wetting, which is then
allocated to baseflow and vaporization. The parameter $V_p$ is the upper limit of the fraction of
wetting returned to the atmosphere as water vapor (Ponce and Shetty, 1995). For the total runoff,
we consider precipitation as the available water competing with evapotranspiration, whose upper
limit is represented by the parameter $U_p$. Similar to $V_p$ in the second stage, $U_p$ can be regarded as
a sort of atmospheric water and energy limit (somewhat analogous to potential evapotranspiration)
and emerges from the interaction of the available energy, vegetation and other catchment
characteristics. To some extent, the MPS model links $Q_s$ and $Q_b$ with $Q$ using $P$ in the first
trade-off and $V_p$ in the second trade-off, so that the forms of different runoff components can be
unified.
Additionally, we compared the distribution of the parameters in the MPS model with that in
Gnann (Gnann et al., 2019) and Siva's work (Sivapalan et al., 2011), which did not omit the
initial abstraction coefficients $\lambda_s$ and $\lambda_b$. There is a very similar spatial pattern of $W_p$ and $V_p$ in
the CONUS. Specifically, high $W_p$ can be seen in the middle of the United States (Great Plains)
and the east (southern parts of the Appalachians) (Figure 7(d)), and high $V_p$ can be seen in the
middle of the United States (Great Plains) and all southern regions (Figure 7(e)). This, to some
extent, illustrates the rationality of the simplification of the original Ponce-Shetty model in
describing the spatial variability of runoff components. According to Ponce and Shetty (1995)



and Sivapalan et al. (2011), the products $\lambda_s W_\mathrm{p}$ and $\lambda_\mathrm{b} V_\mathrm{p}$ are viewed as the initial abstraction to
generate runoff. This definition is reasonable for short-term scales, such as event and annual
scales. However, on the multi-annual scale, the catchment maintains a state of water balance and
water losses can be disregarded (Han et al., 2020). Hence, simplifying $\lambda$ to zero is rational to
quantify and attribute runoff components and offer a new perspective on the long-term catchment
water balance.

**5.3. Uncertainties and Future Improvements**

The sensitivity of runoff to changes in climatic and environmental factors has always been
highly anticipated. Schaake (1990) first introduced the concept of climate elasticity coefficients to
quantify it, defined as the ratio of the relative change in mean annual runoff to the relative change
in climatic factors. Various expressions have been widely applied in evaluating the hydrological
response to multi-annual average climate change (Sun et al., 2014; Xu et al., 2014). The only
climatic factor in the MPS model is $P$, so we primarily focuses on the elasticity of runoff
components to $P$ ($\varepsilon$), which can be expressed as $\varepsilon_{y-P} = \frac{\partial Q_y}{\partial P} / \frac{Q_y}{P}$, quantifying the percentage of
runoff components change caused by 1% change in $P$.
Figure 14 shows elasticities of $Q$, $Q_\mathrm{s}$ and $Q_\mathrm{b}$ to $P$ derived from the MPS model in the CONUS.
We compare the elasticity distribution of the work conducted by Harman et al. (2011), who did
not omit the initial abstraction coefficients $\lambda$. In humid catchments with the aridity index of less
than 1 (such as the west coast and eastern regions of the CONUS), the results from both studies
are very close, with elasticity values from 1 to 2. However, the MPS model noticeably
underestimates the runoff sensitivity to $P$ in semi-arid and arid catchments (such as the Great
Plains). This may be due to the error caused by the assumption that $\lambda$ is a constant when deriving
the MPS model.

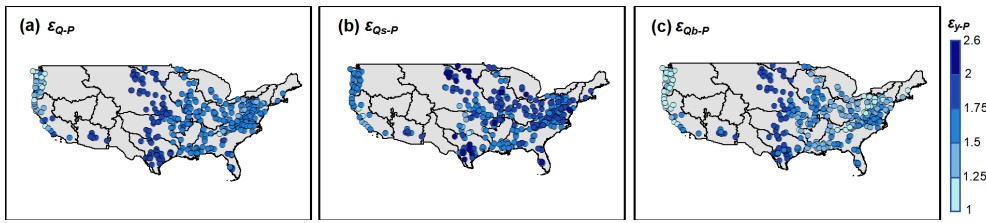


**Figure 14**. The elasticity of (a) total runoff, (b) surface flow and (c) baseflow to precipitation

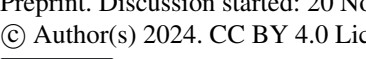



derived the MPS model.

Additionally, the secondary rainfall processes, such as initial abstraction to generate runoff,

precipitation intensity and seasonality should be considered in these regions, which have been
proven to have a significant impact in attribution analysis (He et al., 2022; Ning et al., 2022;
Zhang, 2015). Moreover, the potential evapotranspiration ($E_0$), which indicates the impact of
energy constraints (Huang et al., 2019; Wu et al., 2020), is quite significant in arid and semi-arid
catchments and should be taken into account.

In this paper, we interpret the parameters (i.e., $W_p$, $V_p$ and $U_p$) as a potential upper limit of each

partition stage competing with corresponding runoff components following the annual
Ponce-Shetty model. It is intriguing to discuss whether the connotation of the parameters has
changed from annual to mean annual time scale. On a long-term scale, the initial abstraction
coefficient (i.e., $\lambda_P$ and $\lambda_W$) can be simplified as zero, indicating the loss for generating runoff is
negligible. However, to what extent the initial abstraction coefficient affect precipitation partition
at shorter time scales is still under-determined. The physical and theoretical interpretation of
parameters and their impacts at different time scales are temporarily outside the scope of this
study. However, it is valuable to further research in future work.

The MPS model has only one parameter for controlling each runoff component, which is

arguably simplified but dependent on calibration, and their physical meaning needs further
explanation. We still need to explain the parameters in terms of regional patterns of climatic
and/or catchment attributes, meaning that currently this model can only be applied to gauged
catchments with runoff observations and challenging to transfer to ungauged basins. Cheng et al.
(2022) proposed two machine learning methods to characterize the parameter of the Budyko
framework and further employed them in estimating global runoff partition (Cheng et al., 2023).
Results show that parameters related to vegetation (such as root zone storage capacity, water use
efficiency and vegetation coverage) and climate (such as precipitation depth and climate
seasonality) are the primary controlling factors of the parameter. Similar work can be referred to
(Chen and Ruan, 2023). These investigations provide priori knowledge for quantitatively linking
the parameters of the MPS model to climate forcing and catchment attributes in future work.



## 6. Conclusion

We developed a general formulation (the MPS model) to estimate mean annual runoff components as a function of available water with a synthetic parameter based on a two-stage partition theory, and validated it over 662 catchments across China and the CONUS with further attribution analysis. The concise MPS model provides more accurate runoff components estimation and innovative attribution, offering new insights to long-term water balance and giving additional superiorities toward making predictions of runoff variation under global change. The main conclusions are as follows:

(1) The investigated catchments fit well with the MPS model, with $R^2$ of 0.86, 0.68 and 0.91 for fitting $Q_s$, $Q_b$ and $Q$ in China and $R^2$ of 0.81, 0.44 and 0.80 for fitting $Q_s$, $Q_b$ and $Q$ in the CONUS, implying the MPS model can well reproduce the spatial variability of different runoff components.

(2) The MPS model effectively simulates multi-year runoff components with $R^2$ exceeding 0.97, and the proportion of runoff components relative to precipitation with $R^2$ exceeding 0.94. The spatial distribution of the parameters across China and the CONUS is related to that of climate zoning.

(3) The MPS model has proved effective in quantifying the variations of runoff components induced by precipitation and environmental factors. The estimated and observed $\Delta Q_s$, $\Delta Q_b$ and $\Delta Q$ exhibit high consistency, with an $R^2$ of 0.99 and RMSE of 1.6 mm of $\Delta Q_s$ attribution, $R^2$ of 0.90 and RMSE of 16 mm of $\Delta Q_b$ attribution and $R^2$ of 0.91 and RMSE of 42 mm of $\Delta Q$ attribution, respectively. The variation of $Q_s$ is jointly controlled by $P$ and environmental    factors, while the variation of $Q_b$ is mainly influenced by $V_p$ in most catchments.

In general, this study proposes a general formulation for effectively estimating and attributing the mean annual runoff, surface flow and baseflow. The structure is simple with few parameters and clear physical significance. Its reliability has been authenticated, providing new insights for analyzing watershed water resources in changing environments.


## Author Contribution

Y.H. conceived and designed the study, collected and analyzed the data, and wrote the manuscript. C.L participated in the study design, provided intellectual insights, and reviewed the manuscript for important intellectual content. C.L. and H.Y. guided the research process and critically reviewed the manuscript. All authors have read and approved the final version of the manuscript.

## Competing interests

The authors declare that they have no conflict of interest.

## Acknowledgments

This research was supported by the National Natural Science Foundation of China (grant no. 42041004 and 52309022) and the China National Key R&D Program (grant nos. 2021YFC3000202 and 2022YFC3002802).

## Data and code Availability

The CAMELS data set is available at https://ral.ucar.edu/solutions/products/camels. The hydro-meteorological data of the catchments across China can be obtained from the Zenodo repository via https://zenodo.org/records/11058118 (Li et al., 2024).

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
