# Peer review of "The general formulation for mean annual runoff components estimation and their change attribution"

_Hydrology and Earth System Sciences, 2024_

## Author Comment (AC1)

**Response to reviewers**

We greatly appreciate the reviewers providing valuable and constructive comments on our manuscript. We seriously considered each comment and revised the original manuscript accordingly. The individual comments are replied below. In the following, the reviewer comments are black font and our responses are blue, and the green texts are the quotes of the revised manuscript.

**Reviewer #1**

Greetings. The manuscript entitled "The general formulation for runoff components estimation and attribution at mean annual time scale" with the issue of estimating the various flow components for water resources management purposes. The structure and goals are clear, and the results are consistent with data. This paper can certainly be published after some major adjustments, listed below. I limited the previous revision to the Introduction and Methodology part, I think these need to be fixed before further going down the publication way. These itemized improvements would make the work more scientifically sound and robust. These considerations come from my expertise as a hydrogeologist, so they will pertain to this sphere of competency. Furthermore, I recommend incorporating 'recommended references' and at least having a quick glimpse at 'further reading' for a more precise framing of the work. Best regards.

Reply: We sincerely appreciate your invaluable and constructive suggestions. We have carefully addressed each comment and incorporated corresponding revisions with recommended references into the revised manuscript.

From line 36 on: the description and the classification of these different baseflow components are pretty gross. I understand that the purpose of the work is to categorize all of them as baseflow hydrograph volume portions, but putting in the 'same box' phenomena that are much different from each other doesn't sound good to me. Please discern (from below): deep leakage (if any, if conceptualized); groundwater flow; subsurface (hyporheic) flow; snowmelt. Moreover, these can be caused by highly varying flow sources. We need a strong specification of phenomena and how to consider them here. At least, we should say that there may be geological and climatic (not in the sense of climate change, but yearly-decadal climate cycle) causes.

Groundwater flows and similar ones are related to the local aquifers' geology as the main uncertainty source (see e.g., Schiavo, 2023), while the heterogeneous recharge has a negligible impact (see e.g., D'Oria et al., 2018). Snowmelt is due to yearly-decadal climatic cycles.

Reply: Thank you for your invaluable and professional feedback. We fully agree that categorizing hydrologic processes with distinct origins and mechanisms-such as deep leakage, groundwater flow, subsurface flow, and snowmelt-under the unified term "baseflow" is overly simplistic. In this study, we adopt baseflow as a pragmatic, applied construct: the portion of slow discharge that sustains streamflow during dry periods. We explicitly acknowledge that this aggregate may include groundwater drainage, hyporheic/subsurface exchange, delayed snowmelt, and, where relevant, deeper leakage. Moreover, current large-scale, long-term baseflow separation methods are still unable to distinguish between baseflow contributions from different sources. We acknowledge that the MPS model and baseflow separation methods used in this study cannot reveal internal mechanistic differences among these components. Nevertheless, they are suitable for the macro-scale analysis objectives of this research at the catchment level. Future studies may employ more accurate tracer techniques or modeling approaches to further differentiate these processes.

We have now added a classification of baseflow based on its various origins in the Discussion section (Line 536-553), particularly emphasizing the key driving factors and sources of uncertainty for these different components: "It is important to acknowledge several uncertainties in this study. First, the definition of "baseflow" itself introduces uncertainty. Although widely used as a collective term for delayed streamflow components, baseflow encompasses contributions from hydrologically distinct sources such as groundwater drainage, hyporheic exchange, snowmelt, and deeper subsurface leakage-each with distinct origins, timescales, and sensitivities to environmental factors. For instance, groundwater flow and deep leakage are strongly controlled by geological heterogeneity, including the distribution of rock types, porosity, permeability, faults, and fractures (Schiavo et al., 2023). In contrast, snowmelt baseflow, on the other hand, is mainly driven by temperature variations within interannual to decadal climate cycles.

The definition of baseflow directly influences the selection of catchment areas. Guided by this macro-scale definition-viewing baseflow as the relatively stable portion of total runoff-we included large catchments in our analysis. While this

inclusion may be a source of error, it does not affect the key finding that the MPS model effectively captures the variability of mean annual runoff components across catchments. A sensitivity analysis of the model's performance under different area thresholds is provided in Appendix Table 1. Future studies could combine isotope tracing with hydrological modeling to better quantify the contributions of these different sources".

Table R1 The coefficient of determination ($R^2$) and model parameters for the MPS curve fittings under different area thresholds for selecting catchments in China

| Area thresholds | Number of | $R^2$ | | | Parameters (mm) | | |
|---|---|---|---|---|---|---|---|
| (km$^2$) | catchments | $Q_s$ | $Q_b$ | $Q$ | $W_p$ | $V_p$ | $U_p$ |
| 2,000 | 67 | 0.85 | 0.62 | 0.89 | 3220 | 2794 | 1439 |
| 5,000 | 135 | 0.84 | 0.63 | 0.89 | 3004 | 2651 | 1356 |
| 10,000 | 180 | 0.84 | 0.69 | 0.90 | 3098 | 2614 | 1375 |
| 20,000 | 219 | 0.85 | 0.68 | 0.90 | 3138 | 2585 | 1376 |
| 80,000 | 257 | 0.85 | 0.69 | 0.90 | 3207 | 2487 | 1364 |
| 500,000 | 295 | 0.85 | 0.69 | 0.91 | 3278 | 2428 | 1362 |

As a 'groundwater guy', I usually think that the common ways of defining baseflow from the viewpoint of surface hydrographs partition lack precision (Cheng et al., 2022) or even conceptual correctness (Cartwright et al., 2014).

Reply: We thank the reviewer for this insightful comment. In this study, we defined baseflow as the flow that originates from groundwater and other delayed sources (such as wetlands, lakes, snow and ice), and generally sustains streamflow during dry periods. We agree with you that it lacks precision to separate baseflow from streamflow using a hydrographs partition since the effect from surface water recession is difficult to remove. Therefore, the hydrographs partition or the filtering method only is an approximate to baseflow in theory and application. In previous studies, the filtering method combined with hydrograph analysis are widely used (Beck et al., 2013; Bloomfield et al., 2021; Wang et al., 2021; Xie et al., 2024), some of which have undergone validations in catchments using tracer-based benchmarks (Gonzales et al., 2009; Lott et al., 2016; Wang et al., 2021). Therefore, we think our approach aligns with the pragmatic objectives to estimate mean annual baseflow.

An important point in baseflow estimation is that the structure of the aquifer is not

deterministically achievable; rather than it can be assessed in a Monte Carlo framework. Hence, groundwater baseflow (or, simply, groundwater discharges) should be assessed by achieving multiple realizations upon varying geological conditions (Schiavo, 2023). Where does the role of homogeneous/heterogeneous aquifers may be appraised? At least, one should take the spatial average of the Monte Carlo runs as the most feasible discharge estimation. I think this introductory/discussion point should be incorporated into the work.

Reply: We thank the reviewer for raising this critical point and insight suggestion. We fully agree that accounting for aquifer heterogeneity uncertainty through a Monte Carlo framework would be a more reliable approach. However, it requires much more data and extensive stochastic analysis in up to 662 catchments from both China and USA. In this study, we therefore approached the baseflow using the filtering method and meanwhile added a detailed discussion on this limitation in the manuscript (Section 5.3, Line 554-560): "Second, methodological uncertainty arises from the digital filter method (i.e., the Lyne-Hollick algorithm) for baseflow separation. While practical and widely applied, this approach is deterministic and does not explicitly account for uncertainties related to aquifer heterogeneity, such as spatial variability in hydraulic conductivity, preferential flow paths, or geologic structures. Future work could adopt stochastic frameworks such as Monte Carlo simulation by generating multiple realistic realizations of aquifer heterogeneity to obtain more robust and probabilistic baseflow estimates (Schiavo et al., 2023)".

From line 78 on: one may argue that the aridity index and the estimation of potential evaporation are 'subjective', hence no robust estimations are provided: how to answer this point?

Reply: We appreciate the reviewer's concern that the aridity index $\phi$ might inherit "subjectivity" from estimating potential evaporation. To avoid ambiguity, we explicitly adopt the Penman formulation as our baseline. It is physically based using (radiation, humidity, wind, temperature), has been widely benchmarked and recommended in previous studies (Pimentel et al., 2023; Wang et al., 2025). Because our analyses are conducted at the mean-annual, large-sample scale and our interpretations rely primarily on relative variations and cross-basin gradients in $\phi$, the use of Penman formulation minimizes method-dependent spread and does not affect our qualitative conclusions. We have clarified this choice in the Methods (Line

215-218): We use the Penman equation (Penman, 1948) to estimate $E_0$ of each grid using standard meteorological inputs (e.g., radiation, humidity, wind, temperature). The Penman equation is widely recommended to estimate $E_0$ at catchment scale due to its physical basis (Pimentel et al., 2023; Wang et al., 2025), and it provides a consistent reference for our annual, large-sample analyses.

Table 1. I usually prefer to retrieve parameters from numerical calibration or so. What about the exponent b and the catchment storage capacity? How have they been inferred in the various models? If they are empirically based, do they find any confirmation in numerical applications?

Reply: The shape parameters ($a$, $b$, $c$, $d$) in the equations of Neto et al. (2020) are obtained through an iterative nonlinear calibration procedure. A calibration subset containing half of the total sample size is randomly picked and fitted through a Levenberg-Marquardt nonlinear least squares algorithm, yielding estimates of $a$, $b$, $c$ and $d$. The procedure is repeated 100 times. Mean and standard deviation of the coefficient of determination ($R^2$) between predicted and observed fluxes are calculated for the validation subset, as well as the mean and standard deviation of the fitted parameters. Then, the procedure is repeated for varying values of $(Q_s/P)_{max}$, while its final value is chosen to be the one who yielded the best combined performances for both $Q_s$ and $Q_b$.

Meanwhile, the average soil water storage capacity ($S_p$) is calibrated using an annual-scale Ponce-Shetty model as implemented by Cheng et al. (2021).

I would strongly recommend somehow connecting the baseflow estimations to previous numerical estimations; otherwise, the initial groundwater abstraction 'lambda' indices are pretty vaguely defined. Maybe also the work done by Zuecco et al. (2019) can be helpful.

Reply: We thank the reviewer for the suggestion to connect our baseflow estimates to previous numerical estimations and for pointing us to Zuecco et al. (2019). In the present study, we have chosen a top-down, large-sample hydrological analysis focused on revealing patterns at the mean-annual scale. This approach aligns with our goal of providing a macroscopic overview across diverse catchments. Pursuing detailed numerical modeling (e.g., with MODFLOW) would require site-specific hydrogeological data that are not available for this study. Therefore, while we

acknowledge this as a potential avenue for future site-specific research, we have focused our current work within the stated methodological framework.

The "lambda" abstraction was introduced in the Introduction as a bridge to the groundwater-abstraction literature; it is not used in our analyses.

To better contextualize mechanisms that may affect the slow-flow component, we now expand the Discussion with evidence on subsurface connectivity and its link to stormflow/baseflow behavior, citing Zuecco et al. (2019), who quantified subsurface connectivity and showed its control on event responses and hysteresis patterns in headwater catchments (Line 562-568): "Event-scale analyses indicate that stormflow volumes and hysteresis patterns covary with subsurface connectivity and its timing. For example, Zuecco et al. (2019) who used graph-theory metrics to quantify connectivity in headwater catchments and linked maximum connectivity to stormflow. While our study operates at mean-annual scales, these findings are consistent with our interpretation that geological heterogeneity and preferential pathways (fractures, karst, macropores) modulate the $V_p$ dispersion and, in turn, the aggregate baseflow fraction". This clarifies how connectivity and heterogeneity can modulate the baseflow signal without changing our study scope.

Another major issue is that it has been clear to the scientific community for at least 5 years that groundwater flow is highly spatially heterogeneous, as it is conveyed in preferential pathways where discharges are much higher than elsewhere. Any idea of how to incorporate this viewpoint?

Reply: Thank you for this comment. We acknowledge that explicitly incorporating groundwater heterogeneity would provide deeper mechanistic insight. In response, we have added relevant discussion in Section 5 (Line 560-562): "Additionally, our study did not take into account the spatial heterogeneity of groundwater flow, particularly its preferential pathways through fractures, macropores, or highly permeable sedimentary layers...... Future work could employ numerical models or distributed hydrological models that explicitly represent geological structures to better capture the effects of preferential flow paths at smaller scales ".

**References**

Beck, H.E., van Dijk, A., Miralles, D.G., de Jeu, R.A.M., Bruijnzeel, L.A., McVicar, T.R. &

Schellekens, J. (2013). Global patterns in base flow index and recession based on streamflow observations from 3394 catchments. Water Resources Research, 49(12): 7843-7863. DOI:10.1002/2013wr013918

Bloomfield, J.P., Gong, M., Marchant, B.P., Coxon, G. & Addor, N. (2021). How is Baseflow Index (BFI) impacted by water resource management practices? Hydrology and Earth System Sciences, 25, 5355-5379. DOI: 10.5194/hess-25-5355-2021

Gonzales, A. L., Nonner, J., Heijkers, J. & Uhlenbrook, S. (2009). Comparison of different base flow separation methods in a lowland catchment. Hydrology And Earth System Sciences, 13, 2055-2068. DOI:10.5194/hess-13-2055-2009

Lott, D. A. & Stewart, M. T. (2016). Base flow separation: a comparison of analytical and mass balance methods. Journal of Hydrology, 535, 525-533. DOI:10.1016/j.jhydrol.2016.01.063

Rutledge, A. T (1998). Computer Programs for Describing the Recession of Ground-Water Discharge and for Estimating Mean Ground-Water Recharge and Discharge from Streamflow Records-Update (USGS); https://doi.org/10.3133/wri984148

Wang, Y., Chen, Y. & Chang, H. (2021). Seasonal dynamic identification of Eckhardt digital filter parameters based on isotopes. Water Resources and Hydropower Engineering, 52(12): 99-110. DOI:10.13928/j.cnki.wrahe.2021.12.010

Xie, J., Liu, X., Jasechko, S., Berghuijs, W.R., Wang, K., Liu, C., Reichstein, M., Jung, M. & Koirala, S. (2024). Majority of global river flow sustained by groundwater. Nature Geoscience, 17(8): 770-777. DOI:10.1038/s41561-024-01483-5

---

## Author Comment (AC2)

**Response to reviewers**

We greatly appreciate the reviewers providing valuable and constructive comments on our manuscript. We seriously considered each comment and revised the original manuscript accordingly. The individual comments are replied below. In the following, the reviewer comments are black font and our responses are blue, and the green texts are the quotes of the revised manuscript.

**Reviewer #2**

The manuscript 'The general formulation for runoff components estimation and attribution at mean-annual time scale' proposes a concise MPS framework for partitioning total runoff into surface and baseflow components. The topic is timely and the presentation generally clear. With several focused revisions—mainly on scope, definitions, robustness checks, and uncertainty, the paper will be suitable for publication.

Reply: We sincerely appreciate your constructive suggestions. We have carefully addressed each comment and incorporated corresponding revisions into the revised manuscript.

Concerns:

Please explicitly delimit applicability to small/medium catchments and justify the exclusion (or stratified analysis) of large basins, where digital filters can misclassify delayed stormflow as baseflow. Provide a short area-threshold sensitivity (e.g., ≤500/1,000/2,500/5,000 km²) showing effects on BFI and on MPS fits; discuss implications for scaling to large rivers (cf. recent global assessments).

Reply: Thank you for this important comment. According to Xie et al. (2024), the underlying assumptions of digital filter baseflow separation methods may not be appropriate for large basins. For example, headwater stormflow of large basins may take weeks to reach the basin outlet and become the low-frequency component of downstream flow. Consequently, these separation methods typically overestimate baseflow in large basins because they misidentify upstream stormflow as baseflow (Rutledge, 1998). Therefore, we focus our analysis on small and medium catchments with an area≤500,000 km² to minimize the influence of channel routing.

Furthermore, we conducted the area-threshold sensitivity analysis in China as

recommended. We systematically tested the effects of varying area thresholds on the performance of the fitted MPS curves. The results showed that the goodness of fit for the MPS relationships remained robust and did not exhibit significant degradation across these different area thresholds (Table A1). We interpret the stability of the MPS fits to mean that the functional relationship between available water and runoff components (as captured by the MPS model) may be scale-invariant within the range of basin sizes studied.

Table A1 The coefficient of determination ($R^2$) and model parameters for the MPS curve fittings under different area thresholds for selecting catchments in China

| Area thresholds (km$^2$) | Number of catchments | $R^2$ | | | Parameters (mm) | | |
|---|---|---|---|---|---|---|---|
| | | $Q_s$ | $Q_b$ | $Q$ | $W_p$ | $V_p$ | $U_p$ |
| 2,000 | 67 | 0.85 | 0.62 | 0.89 | 3220 | 2794 | 1439 |
| 5,000 | 135 | 0.84 | 0.63 | 0.89 | 3004 | 2651 | 1356 |
| 10,000 | 180 | 0.84 | 0.69 | 0.90 | 3098 | 2614 | 1375 |
| 20,000 | 219 | 0.85 | 0.68 | 0.90 | 3138 | 2585 | 1376 |
| 80,000 | 257 | 0.85 | 0.69 | 0.90 | 3207 | 2487 | 1364 |
| 500,000 | 295 | 0.85 | 0.69 | 0.91 | 3278 | 2428 | 1362 |

Clarify the boundary between baseflow/slow flow and surface/fast flow. At minimum, acknowledge that baseflow aggregates multiple processes (groundwater discharge, hyporheic/subsurface flow, delayed snowmelt, and—if relevant—deep leakage).

Reply: Thank you for this suggestion. We acknowledge that the term "baseflow" aggregates multiple delayed flow processes, including groundwater discharge, hyporheic exchange, subsurface stormflow, delayed snowmelt, and deep leakage with distinct origins, timescales and physical mechanisms. In response, we have expanded the Discussion section (Line 536-553) to explicitly recognize that baseflow represents an integrated concept encompassing these heterogeneous components: "It is important to acknowledge several uncertainties in this study. First, the definition of "baseflow" itself introduces uncertainty. Although widely used as a collective term for delayed streamflow components, baseflow encompasses contributions from hydrologically distinct sources such as groundwater drainage, hyporehic exchange, snowmelt, and deeper subsurface leakage-each with distinct origins, timescales, and sensitivities to environmental factors. For instance, groundwater flow and deep leakage are strongly

controlled by geological heterogeneity, including the distribution of rock types, porosity, permeability, faults, and fractures (Schiavo et al., 2023). In contrast, snowmelt baseflow, on the other hand, is mainly driven by temperature variations within interannual to decadal climate cycles. Future studies could combine isotope tracing with hydrological modeling to better quantify the contributions of these different sources".

Strengthen the interpretation of Wp, Vp, and Up: (1) outline hypothesized controls (soil/rock properties, storage capacity, seasonality); (2) report basic identifiability/collinearity checks (against Budyko-type indices); (3) add a cross-region transfer test (China→CONUS and vice versa) to show portability; (4) explain how to calculate the changes in parameters when attributing the variations in runoff components, such as ΔUp.

Reply: We thank for these insightful suggestions.

(1) We agree that a clearer physical interpretation of the parameters is beneficial. In the revised Discussion section, we have added the following paragraph: "$W_p$ is influenced by soil properties and available storage capacity, determining the fraction of precipitation that rapidly becomes surface runoff versus what is stored (Line 503-504)"; "The parameter $V_p$ is the upper limit of the fraction of wetting returned to the atmosphere as water vapor (Ponce and Shetty, 1995), and is likely responds to subsurface characteristics such as aquifer permeability and geological layering (506-508)".

(2) We have compared the results with Budyko equations in Section 5.1.

(3) In the doctoral thesis of the first author (He, 2025), the explicit equations relating the parameters ($W_p$, $V_p$ and $U_p$) to catchment attributes (e.g., rainfall intensity, snow fraction, topographic indices, elevation, permeability) have been established using a large dataset of Chinese catchments. These relationships have been validated within China and shown to provide reliable runoff components estimates for ungauged catchments. While a direct cross-region transfer test (e.g., China → CONUS) is beyond the scope of this paper, the attribute-based parameterization approach provides a strong foundation for geographical generalizability. We will explicitly recommend and undertake this important validation in future work.

(4) The changes in parameters between two periods (e.g., $\Delta U_p$) are calculated as follows: First, the $U_{p1}$ and $U_{p2}$ are inversely estimated from the observed total runoff

using Equation (14) for period 1 and period 2, respectively. Then, the change of $U_p$ is computed simply as the difference between two periods ($\Delta U_p = U_{p2} - U_{p1}$). Similarly, $\Delta P$ represents the change in mean annual precipitation between the two periods. These derived changes ($\Delta P$, $\Delta U_p$) are then used in the attribution framework (Equation 17(b)) to quantity variations in total runoff to climatic and environment changes.

Discuss how known aquifer heterogeneity and preferential flow may map onto parameter dispersion (notably Vp).

Reply: Thank you for this important comment. In response, we have added the following discussion to Section 5 (Line 507-514) of the revised manuscript: "..., and is likely responds to subsurface characteristics such as aquifer permeability and geological layering. For instance, in highly heterogeneous aquifers with well-developed preferential pathways (e.g., fractured rock or karst systems), water is rapidly drained toward the stream, leading to a higher efficiency of baseflow production and thus a lower $V_p$ value (as less water is retained for evaporation). Conversely, in catchments with more homogeneous, porous media (e.g., sandy aquifers), water movement is slower and more diffuse, potentially allowing for a greater fraction of stored water to be evaporated, resulting in a higher $V_p$".

Minor comments

Unify color scales/units; add 95% confidence bands to CDF/scatter plots.

Reply: Done.

Provide a concise symbol table (first occurrence) and standardize terminology ('runoff components' vs 'flow components'; 'baseflow/slow flow').

Reply: We thank the reviewer for the suggestion regarding terminology and symbols. We have thoroughly reviewed the manuscript to ensure standardized terminology. The terms "runoff components" and "baseflow" are now used consistently throughout the text.

Regarding the symbol table, we have defined each symbol upon its first occurrence in the text. We believe this approach provides clarity to readers without a symbol table.

Briefly document missing-data criteria, period lengths by region, and QC steps.

Reply: Thank you for this important suggestion. We have supplemented the catchment screening criteria in Section 3.1, with detailed procedures available in He et al. (2025).

In Table 1, state whether exponents/capacities are calibrated or empirical and, where possible, cite numerical/observational backing.

Reply: We have added the sources of parameters in Table 1.

Add a short analysis or paragraph on precipitation seasonality effects on BFI and on partitioning assumptions at the annual scale.

Reply: Thank you for this comment. We have added some discussion in Line 604-609: "In addition, the seasonality of precipitation measures the concentration of precipitation within a year. The more concentrated the precipitation, the more likely it is to generate surface runoff, resulting in greater intra-annual fluctuations in the BFI and a lower annual BFI. In contrast, in catchments with evenly distributed precipitation, soil water and groundwater are replenished consistently and gradually, leading to relatively stable intra-annual BFI and a higher annual BFI".

For the phrase 'As for $\Delta Q$ attribution' on line 394, perhaps 'attribution' should be removed.

Reply: Done.

**References**

Beck, H.E., van Dijk, A., Miralles, D.G., de Jeu, R.A.M., Bruijnzeel, L.A., McVicar, T.R. & Schellekens, J. (2013). Global patterns in base flow index and recession based on streamflow observations from 3394 catchments. Water Resources Research, 49(12): 7843-7863. DOI:10.1002/2013wr013918

Bloomfield, J.P., Gong, M., Marchant, B.P., Coxon, G. & Addor, N. (2021). How is Baseflow Index (BFI) impacted by water resource management practices? Hydrology and Earth System Sciences, 25, 5355-5379. DOI: 10.5194/hess-25-5355-2021

Gonzales, A. L., Nonner, J., Heijkers, J. & Uhlenbrook, S. (2009). Comparison of different base flow separation methods in a lowland catchment. Hydrology And Earth System Sciences, 13, 2055-2068. DOI:10.5194/hess-13-2055-2009

Lott, D. A. & Stewart, M. T. (2016). Base flow separation: a comparison of analytical and mass balance methods. Journal of Hydrology, 535, 525-533. DOI:10.1016/j.jhydrol.2016.01.063

Rutledge, A. T (1998). Computer Programs for Describing the Recession of Ground-Water Discharge and for Estimating Mean Ground-Water Recharge and Discharge from Streamflow Records-Update (USGS); https://doi.org/10.3133/wri984148

Wang, Y., Chen, Y. & Chang, H. (2021). Seasonal dynamic identification of Eckhardt digital filter parameters based on isotopes. Water Resources and Hydropower Engineering, 52(12): 99-110. DOI:10.13928/j.cnki.wrahe.2021.12.010

Xie, J., Liu, X., Jasechko, S., Berghuijs, W.R., Wang, K., Liu, C., Reichstein, M., Jung, M. & Koirala, S. (2024). Majority of global river flow sustained by groundwater. Nature Geoscience, 17(8): 770-777. DOI:10.1038/s41561-024-01483-5